# Impact of Reappraisal of Fluoroquinolone Minimum Inhibitory Concentration Susceptibility Breakpoints in Gram-Negative Bloodstream Isolates

**DOI:** 10.3390/antibiotics9040189

**Published:** 2020-04-17

**Authors:** Stephanie C. Shealy, Matthew M. Brigmon, Julie Ann Justo, P. Brandon Bookstaver, Joseph Kohn, Majdi N. Al-Hasan

**Affiliations:** 1Department of Pharmacy, Prisma Health Richland Hospital, Columbia, SC 29203, USA; shealysc@email.sc.edu (S.C.S.); justoj@cop.sc.edu (J.A.J.); bookstaver@cop.sc.edu (P.B.B.); joseph.kohn@prismahealth.org (J.K.); 2Department of Clinical Pharmacy and Outcomes Sciences, University of South Carolina College of Pharmacy, Columbia, SC 29208, USA; 3Department of Medicine, Baylor Scott and White, Texas A&M Health Science Center College of Medicine, Temple, TX 76502, USA; matthew.brigmon@bswhealth.org; 4Department of Medicine, University of South Carolina School of Medicine, Columbia, SC 29209, USA; 5Prisma Health-Midlands, Columbia, SC 29203, USA

**Keywords:** sepsis, bacteremia, ciprofloxacin, chemotherapeutics, *Escherichia coli*, non-fermenters/*Pseudomonas aeruginosa*

## Abstract

The Clinical Laboratory Standards Institute lowered the fluoroquinolone minimum inhibitory concentration (MIC) susceptibility breakpoints for *Enterobacteriaceae* and glucose non-fermenting Gram-negative bacilli in January 2019. This retrospective cohort study describes the impact of this reappraisal on ciprofloxacin susceptibility overall and in patients with risk factors for antimicrobial resistance. Gram-negative bloodstream isolates collected from hospitalized adults at Prisma Health-Midlands hospitals in South Carolina, USA, from January 2010 to December 2014 were included. Matched pairs mean difference (MD) with 95% confidence intervals (CI) were calculated to examine the change in ciprofloxacin susceptibility after MIC breakpoint reappraisal. Susceptibility of *Enterobacteriaceae* to ciprofloxacin declined by 5.2% (95% CI: −6.6, −3.8; *p* < 0.001) after reappraisal. The largest impact was demonstrated among *Pseudomonas aeruginosa* bloodstream isolates (MD −7.8, 95% CI: −14.6, −1.1; *p* = 0.02) despite more conservative revision in ciprofloxacin MIC breakpoints. Among antimicrobial resistance risk factors, fluoroquinolone exposure within the previous 90 days was associated with the largest change in ciprofloxacin susceptibility (MD −9.3, 95% CI: −16.1, −2.6; *p* = 0.007). Reappraisal of fluoroquinolone MIC breakpoints has a variable impact on the susceptibility of bloodstream isolates by microbiology and patient population. Healthcare systems should be vigilant to systematically adopt this updated recommendation in order to optimize antimicrobial therapy in patients with bloodstream and other serious infections.

## 1. Introduction

Despite several Food and Drug Administration (FDA) safety alerts since 2008, fluoroquinolones remain the third most commonly prescribed class of antibiotics in the United States [1]. Fluoroquinolones are attractive antimicrobial options for certain indications due to their broad spectrum of activity and high bioavailability of oral formulations [2,3,4]. Resistance to fluoroquinolones among *Enterobacteriaceae* and non-fermenters (bacteria that cannot catabolize glucose, such as *Pseudomonas aeruginosa* and *Acinetobacter baumannii*) is well described, and may pose increased risk of mortality and significant burden to the healthcare system [5,6]. Furthermore, elevated fluoroquinolone minimum inhibitory concentration (MIC) values among Gram-negative isolates have been associated with poor clinical outcomes, such as longer duration of hospital stay and treatment failures, despite meeting the definition for susceptibility [7,8].

In 2015, the United States Committee on Antimicrobial Susceptibility Testing (USCAST) proposed lowering the ciprofloxacin MIC breakpoint value for *Enterobacteriaceae* to ≤0.25 mcg/mL and for non-fermenters to ≤0.5 mcg/mL. At the time, the Clinical Laboratory Standards Institute (CLSI) and the FDA endorsed a ciprofloxacin MIC breakpoint value for *Enterobacteriaceae* and non-fermenters of ≤1 mcg/mL. This proposed reappraisal in the MIC breakpoint followed an acknowledgment of a trend of declining susceptibility among *Enterobacteriaceae* to fluoroquinolones. Specifically, the susceptibility of *Enterobacteriaceae* to ciprofloxacin declined from 95.2% to 81.1% between 1998 and 2013. The committee noted that the same declining trend in ciprofloxacin susceptibility was not demonstrated among *P. aeruginosa* isolates, with relatively consistent susceptibility rates since drug approval of approximately 70%. Further evidence to support the reappraisal in MIC breakpoint values was provided by pharmacokinetic-pharmacodynamic (PK-PD) target attainment analyses conducted by USCAST. Among *Enterobacteriaceae*, the probability of PK-PD target attainment (defined as 1 log10 CFU reduction from baseline) with the ciprofloxacin 400 mg intravenously every 8 h regimen was 99.4% and 82.3% for organisms with MIC values of 0.25 and 0.5 mcg/mL, respectively. Among *P. aeruginosa* isolates, the probability of PK-PD target attainment with this ciprofloxacin regimen was 97.2% and 61.9% for organisms with the MIC values of 0.5 and 1 mcg/mL, respectively [9].

In response to this evidence, the CLSI and FDA updated the MIC breakpoint values and associated interpretive criteria for *Enterobacteriaceae* and non-fermenters in January 2019 to align with the USCAST recommendation. This study aims to characterize the impact of reappraisal of MIC breakpoint values on ciprofloxacin susceptibility among Gram-negative bloodstream isolates [10,11].

## 2. Results

### 2.1. Microbiology

A total of 1055 Gram-negative bloodstream isolates were identified over the five year study period, including 967 (92%) *Enterobacteriaceae* and 88 (8%) non-fermenters. *Escherichia coli* was the most common organism identified (51%), followed by *Klebsiella spp* (19%). The most common non-fermenter identified was *Pseudomonas aeruginosa*, which comprised 73% of the group and 6% overall (Figure 1).

### 2.2. MIC Distribution by Organism

Among *Enterobacteriaceae*, 83% and 78% of isolates demonstrated a ciprofloxacin MIC value ≤1 and ≤0.25 mcg/mL, respectively, indicating an overall mean difference in ciprofloxacin susceptibility by previous and current breakpoints of 5%. For non-fermenters, 89% and 81% of isolates demonstrated a ciprofloxacin MIC value of ≤1 mcg/mL and ≤0.5 mcg/mL, respectively, reflecting an overall mean difference in ciprofloxacin susceptibility by previous and current breakpoints of 8%. An additional 10% of non-fermenters had an MIC of 0.5 mcg/mL, which is still considered susceptible based on the new breakpoints criteria (Figure 2). By the updated MIC breakpoint of ≤0.25 mcg/mL, *E. coli* demonstrated the lowest susceptibility to ciprofloxacin (71%), while *Enterobacter spp.* demonstrated the highest (94%), as demonstrated in Table 1.

With consideration of reduced breakpoint for *Enterobacteriaceae* from ≤1 to ≤0.25 mcg/mL, *E. coli* and *Klebsiella spp.* demonstrated statistically significant changes in ciprofloxacin susceptibility by −5.5% (95% CI: −7.5, −3.6; *p* < 0.001) and −5.4% (95% CI: −8.5, −2.3; *p* < 0.001), respectively. In comparison, the mean difference in ciprofloxacin susceptibility was −7.8% (95% CI: −14.6, −1.1; *p* = 0.02) for *P. aeruginosa* isolates by the reduced breakpoint from ≤1 to ≤0.5 mcg/mL (Table 1).

### 2.3. Impact of Reappraisal of MIC Values among Patients with Resistance Risk Factors

Among patients with bloodstream infections due to *Enterobacteriaceae*, those with certain risk factors for fluoroquinolone resistance had a significant decline in ciprofloxacin susceptibility with a reappraisal of the MIC breakpoints (Table 2 and Figure 3) The largest impact was demonstrated among patients with fluoroquinolone use within the previous 90 days, followed by residence in a skilled nursing facility. Risk factors that did not have a statistically significant impact on the mean difference in ciprofloxacin susceptibility were recent outpatient procedures and fluoroquinolone use within the prior 90–180 days. Patients without fluoroquinolone resistance risk factors experienced a statistically significant decline in ciprofloxacin susceptibility, although to a lesser extent when compared to the entire population with bloodstream infections caused by *Enterobactriaceae* (3.8% vs. 5.2%).

## 3. Discussion

### 3.1. Impact of Reappraisal of MIC Breakpoints

Reappraisal of the MIC breakpoint values for *Enterobacteriaceae* and non-fermenters by CLSI and FDA in 2019 has a variable impact on ciprofloxacin susceptibility among Gram-negative bloodstream isolates. Among the five most common gram-negative bloodstream isolates, *P. aeruginosa* had the greatest impact, despite a more conservative reappraisal of MIC breakpoints from ≤1 to ≤0.5 mcg/mL. Of all non-fermenting isolates, 8% met susceptibility based on previous breakpoints and were considered non-susceptible by updated breakpoints. This represents a significant decline in ciprofloxacin susceptibility. With dynamic and variable resistance mechanisms, *P. aeruginosa* is a highly virulent pathogen that requires special consideration to optimize antimicrobial therapy. The consequences of inadequate antimicrobial therapy include the progression of infection and potentiation of further resistance. Reappraisal of the MIC breakpoint value for fluoroquinolones may aid in the optimization of antimicrobial therapy for a significant proportion of patients with bloodstream infections caused by *P. aeruginosa*. The magnitude of the change in ciprofloxacin susceptibility after the reappraisal of breakpoints was lower among *Enterobacteriaceae* bloodstream isolates, a 5% decline overall despite reducing breakpoints from ≤1 to ≤0.25 mcg/mL. Among *Enterobacteriaceae*, *Enterobacter* spp. has the least impact of lowering susceptibility breakpoints.

The study also characterized the variability in the impact of breakpoint reappraisal based on risk factors for fluoroquinolone resistance in patients with *Enterobacteriaceae* bloodstream infections. The greatest impact observed was among bloodstream isolates from patients who had fluoroquinolone exposure within the past 90 days. Of these isolates, 9.3% met susceptibility based on previous MIC breakpoints were considered non-susceptible by updated breakpoints. Residence at skilled nursing facilities demonstrated a similar, significant impact on ciprofloxacin susceptibility with a mean difference of 7.9%. These two risk factors are considered major contributors to fluoroquinolone resistance and have been incorporated into clinical risk scoring tools for optimization of empirical antimicrobial selection in this patient population [12]. While other risk factors, such as fluoroquinolone exposure within 90–180 days and recent outpatient procedures, demonstrated a decline in ciprofloxacin susceptibility by updated breakpoints, this finding was not statistically significant, likely due to relatively small number of patients with these two risk factors. Notably, ciprofloxacin susceptibility declined by only 3.8% in patients without risk factors for fluoroquinolone resistance.

### 3.2. Systematic Adoption of Updated MIC Breakpoint Values

It is imperative to deploy systematic adoption of these updated MIC breakpoint values in order to positively impact patient care. This effort, as any effort in response to updated interpretive values or other laboratory standards, should involve key microbiology laboratory personnel, members of the antimicrobial stewardship program, specialized clinicians, information technologists, and administration. Without an organized effort to educate and update technology to reflect updated breakpoints, the impact of the reappraisal of MIC breakpoint values for *Enterobacteriaceae* and non-fermenters may not reach the level of patient care due to the reliance of prescribers on the interpretive values assigned in the microbiology reports. Barriers to systematic adoption may include lack of easily modifiable electronic infrastructure, lack of intangible resources (i.e., time), and perceived unimportance. Findings such as those described in this study should help support the importance of an urgent effort to adopt the reappraisal of MIC breakpoints. In the absence of an organized effort to adopt the updated MIC breakpoints, specialized clinicians must be vigilant to educate prescribers on the updates and provide context to the associated interpretive values. Members of the antimicrobial stewardship team are well-positioned to educate prescribers at general and patient-specific levels.

### 3.3. Impact on Fluoroquinlone Prescribing

Reappraisal of fluoroquinolone susceptibility breakpoints is unlikely to influence the empirical use of fluoroquinolones for Gram-negative bloodstream infections. The non-stratified use of empirical fluoroquinolones is discouraged due to already high overall fluoroquinolone resistance rates among community-acquired and healthcare-associated Gram-negative bloodstream isolates [13,14]. For the most part, intravenous broad-spectrum beta-lactams remain the first-line agents for empirical therapy of Gram-negative bloodstream infections in hospitalized patients without major allergic reactions [15].

However, the downward shift in fluoroquinolone susceptibilities after breakpoint reappraisal will have an impact on targeted therapy of Gram-negative bloodstream infections, particularly intravenous to oral switch options. Currently, 70–74% of patients with Gram-negative bloodstream infections receive oral fluoroquinolones upon transition from intravenous to oral antimicrobial therapy [16,17]. This likely constitutes the majority of patients with bloodstream infections due to fluoroquinolone-susceptible Gram-negative bacilli given overall 25% fluoroquinolone resistance rates [5,18]. Based on the results of the current study, lowering fluoroquinolone susceptibility breakpoints implies that an additional 5–8% of patients may not have effective and highly bioavailable oral antimicrobial options for treatment of Gram-negative bloodstream infections [2,19,20]. Future clinical studies should determine whether optimization of oral beta-lactam dosing may improve target attainment and, hence, the clinical outcomes of Gram-negative bloodstream infections in these patients. This also creates a niche for drug development of novel oral antimicrobials that may fill the gap created by declining fluoroquinolone susceptibilities and recent safety concerns.

### 3.4. Strengths and Limitations

The large sample size of *Enterobacteriaceae* bloodstream isolates represents the major strength in this study. Examining the change in fluoroquinolone susceptibilities by the presence of major risk factors for resistance adds novelty to this work. The study also has limitations. First, data were derived from two community hospitals within the same healthcare system and geographical area. This may limit generalizability to other settings with different hospital epidemiology and antimicrobial resistance rates. Second, recent studies have demonstrated a decline in fluoroquinolone use after the FDA safety warnings [21]. This decline in prescription rates may influence fluoroquinolone resistance rates in the future.

## 4. Materials and Methods

### 4.1. Setting

This study took place at Prisma Health Richland and Prisma Health Baptist hospitals (formerly Palmetto Health). Combined, these hospitals are comprised of 1100 beds and provide a variety of medical, surgical, and subspecialty services to patients mostly residing in the Midlands region of South Carolina.

### 4.2. Study Design and Definitions

This was a retrospective cohort study that included Gram-negative bloodstream isolates identified over a five-year period (1 January 2010 through 31 December 2014). Gram-negative bloodstream isolate was defined as the growth of any aerobic Gram-negative bacillus in blood culture. Isolates were identified through the Prisma Health microbiology database. Within this central laboratory, genus and species were identified using matrix-assisted laser desorption ionization time of flight mass spectrometry (MALDI–TOF). MICs were determined using Vitek^®^ automated antimicrobial sensitivity testing instruments. Specific antimicrobial susceptibility testing previously identified as unreliable using this instrument through quality assurance was verified with disk diffusion and/or Kirby–Bauer methods. Bloodstream isolates from hospitalized adults with first episodes of Gram-negative bloodstream infection were included. Populations that were excluded were children <18 years, patients with recurrent bloodstream infection, patients with polymicrobial blood cultures, and patients treated in the outpatient setting.

Previous MIC breakpoint values and associated interpretive designations were defined as those values assigned by CLSI and FDA immediately prior to reappraisal in January 2019 (most recently described in the 28th edition of CLSI Document M100). Updated MIC breakpoint values and associated interpretive designations were defined as those values assigned by CLSI and FDA in January 2019 (first described in the 29th edition of CLSI Document M100). Risk factors for fluoroquinolone resistance among *Enterobacteriaceae* were identified through a previously described case-control study conducted in this patient population [12].

### 4.3. Statistical Analysis

Descriptive statistics were used to characterize microbiology and the MIC distributions. Matched pairs mean difference (MD) with 95% confidence intervals was calculated to examine the change in ciprofloxacin susceptibility after the reappraisal.

## 5. Conclusions

Reappraisal of MIC susceptibility breakpoint values for *Enterobacteriaceae* and non-fermenters has a variable impact on the interpretive designation for ciprofloxacin susceptibility. *P. aeruginosa* is the most impacted organism, and patients with fluoroquinolone use within the previous 90 days are the most impacted risk factor group. Our data also indicate a possible benefit from an antimicrobial stewardship perspective in nursing home residents or those who have had recent fluoroquinolone exposure. Timely implementation of new fluoroquinolone susceptibility breakpoints by local microbiology laboratories and healthcare systems is crucial for the optimization of antimicrobial therapy in patients with bloodstream and other serious infections.

## Figures and Tables

**Figure 1 antibiotics-09-00189-f001:**
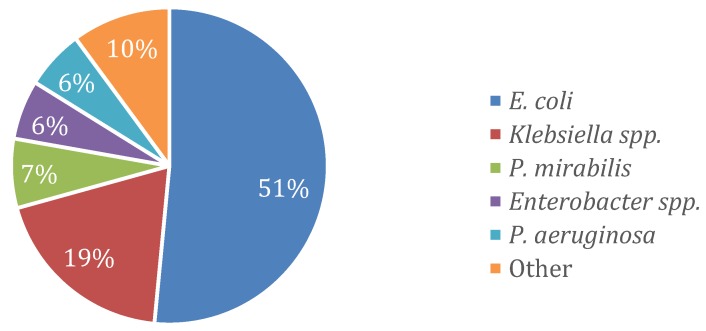
Microbiology of Gram-negative bloodstream isolates.

**Figure 2 antibiotics-09-00189-f002:**
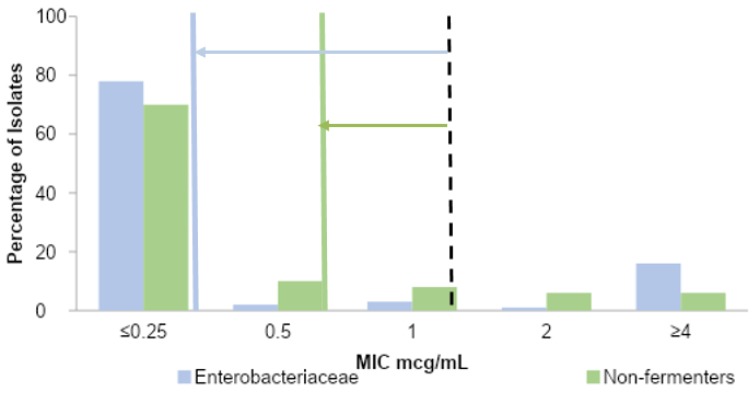
Ciprofloxacin minimum inhibitory concentration (MIC) values for *Enterobacteriaceae* and non-fermenters. Dashed line represents previous ciprofloxacin MIC breakpoints for *Enterobacteriaceae* and non-fermenters, and the solid lines represent updated ciprofloxacin MIC breakpoints for *Enterobacteriaceae* (in blue) and non-fermenters (in green). The arrows reflect the magnitude of the MIC shift for *Enterobacteriaceae* (in blue) and non-fermenters (in green).

**Figure 3 antibiotics-09-00189-f003:**
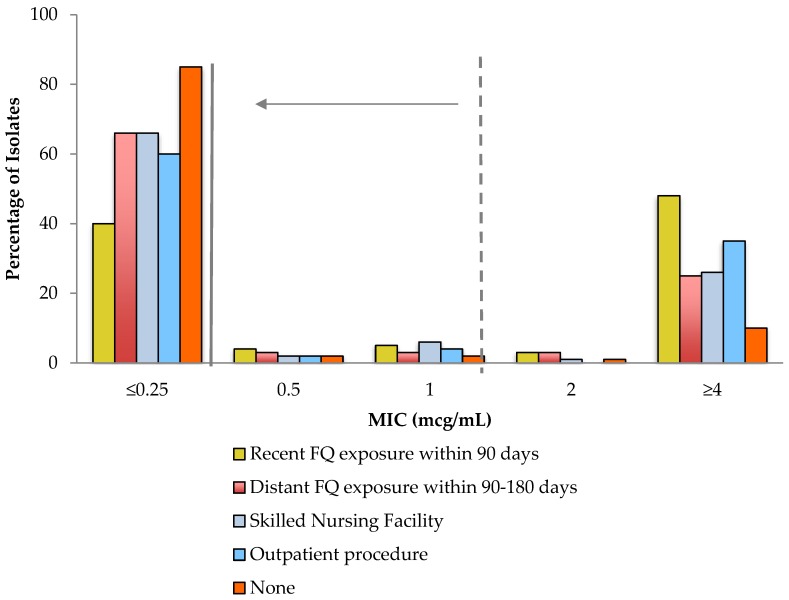
Ciprofloxacin MIC values for *Enterobacteriaceae* by risk factors for resistance. The dashed line represents the previous breakpoint and the solid line represents the updated breakpoint. FQ: fluoroquinolone.

**Table 1 antibiotics-09-00189-t001:** Mean difference in ciprofloxacin susceptibilities by previous and updated breakpoints.

Organism	N	Previous Susceptibility	Updated Susceptibility	Mean Difference (95% CI)	*p*-Value
*Enterobacteriaceae*	967	803 (83.0)	753 (77.9)	−5.2 (−6.6, −3.8)	< 0.001
*E. coli*	543	416 (76.6)	386 (71.1)	−5.5 (−7.5, −3.6)	< 0.001
*Klebsiella spp.*	205	187 (91.2)	176 (85.9)	−5.4 (−8.5, −2.3)	< 0.001
*P. mirabilis*	72	63 (87.5)	60 (83.3)	−4.2 (−8.9, 0.6)	0.08
*Enterobacter spp.*	67	65 (97.0)	63 (94.0)	−3.0 (−7.2, 1.2)	0.16
Non-fermenters	88	78 (88.6)	71 (80.7)	−8.0 (−13.7, −2.2)	0.007
*P. aeruginosa*	64	57 (89.1)	52 (81.3)	−7.8 (−14.6, −1.1)	0.02

CI: confidence interval.

**Table 2 antibiotics-09-00189-t002:** Mean difference in ciprofloxacin susceptibilities for *Enterobacteriaceae* by previous and updated breakpoints based on risk factors for fluoroquinolone resistance.

Risk Factor	N *	Previous Susceptibility	Updated Susceptibility	Mean Difference (95% CI)	*p*-Value
Fluoroquinolone use within prior 90 d	75	37 (49.3)	30 (40.0)	−9.3 (−16.1, −2.6)	0.007
Fluoroquinolone use within prior 90–180 d	32	23 (71.9)	21 (65.6)	−6.3 (−15.1, 2.6)	0.16
Residence at skilled nursing facility	152	112 (73.7)	100 (65.8)	−7.9 (−12.2, −3.6)	< 0.001
Outpatient GI/GU procedure within prior 30 d	55	36 (65.5)	33 (60.0)	−5.5 (−11.7, 0.7)	0.08
None	682	607 (89.0)	581 (85.2)	−3.8 (−5.3, −2.4)	< 0.001

CI: confidence interval; GI: gastrointestinal; GU: genitourinary. * Patients may have multiple risk factors for fluoroquinolone resistance.

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
