# Peer review of "Impact of Reappraisal of Fluoroquinolone Minimum Inhibitory Concentration Susceptibility Breakpoints in Gram-Negative Bloodstream Isolates"

_antibiotics, 2020, doi:10.3390/antibiotics9040189_

Round 1

Reviewer 1 Report

This study is an important additive to the literature on antimicrobial resistance.  As we wait for the AST platforms to catch up with the latest CLSI guidance, it is important for us to understand the impact on our antibiogram in order to shape our antimicrobial stewardship interventions.

A few comments:

While most of the readers will know what you mean by "non-fermenters", I did not see where you defined the term non-fermenter.

Figures 2 and 3 are interesting and colorful and show me the specific groups or number of isolates that the updated breakpoints will be now moving into "resistant" categories, but not sure they add much to Table 1.  I assume the point of these figures are to show which isolates are getting dropped with the new breakpoints, but not sure if this is helpful and this particular point is not highlighted in the discussions

I could say the same for figure 4 but if you elect to keep it, note that there is a red bar with out a legend, presumably it is the "none" risk factor category.

Regarding the risk factor discussion:  I believe i understand what you are trying to say, specifically, those who received fluoroquinolones in the previous 90 days and those residing in SNF will be impacted most by the new breakpoints.  However the way it is framed in the manuscript implies that the risk factors for resistance are the reason for the difference in breakpoint changes.  For example, the title of the section is "Impact of Resistance Factors on MIC", and while the described risk factors may have an impact on the MIC, this is not actually what you are showing with your study.  You are showing that the patients with these risk factors will be affected most with the new breakpoints.  While the breakpoint changes are independent of the risk factors you describe, because the breakpoints were already decided upon, it is still a notable finding that there are specific groups that will be impacted the most with these updated breakpoints.  These are the groups that we will be able to target our Stewardship interventions first.

Lastly, the last sentence in Line 229 thru 231, seems to belong more in the discussion section. 

Author Response

This study is an important additive to the literature on antimicrobial resistance.  As we wait for the AST platforms to catch up with the latest CLSI guidance, it is important for us to understand the impact on our antibiogram in order to shape our antimicrobial stewardship interventions.

A few comments:

While most of the readers will know what you mean by "non-fermenters", I did not see where you defined the term non-fermenter.

  • A definition was added on Page 1, line 44 (“bacteria which cannot catabolize glucoses”)

Figures 2 and 3 are interesting and colorful and show me the specific groups or number of isolates that the updated breakpoints will be now moving into "resistant" categories, but not sure they add much to Table 1.  I assume the point of these figures are to show which isolates are getting dropped with the new breakpoints, but not sure if this is helpful and this particular point is not highlighted in the discussions

  • Thank you for this comment. It provoked good and detailed discussion among the authors. For figure 3, we agree this did not provide any information that was additional to table 1, so figure 3 was removed and the previous figure labeled as figure 4 is now labeled as figure 3. For figure 2, we actually believe this figure provides additional information to table 1 in regards to the proportion of isolates that had specific MIC values (0.5mcg/mL vs. 1mcg/mL) which we think is important information for the paper. Some additional context was also added to the text to give further rationale for the inclusion of figure 2 (Lines 85-87)

I could say the same for figure 4 but if you elect to keep it, note that there is a red bar without a legend, presumably it is the "none" risk factor category.

  • Thank you for this comment. The legend has been updated, red to reflect “none” risk factor category. The rationale for keeping figure 4 (now labeled as figure 3), is the same as mentioned above for figure 2.

Regarding the risk factor discussion:  I believe i understand what you are trying to say, specifically, those who received fluoroquinolones in the previous 90 days and those residing in SNF will be impacted most by the new breakpoints.  However the way it is framed in the manuscript implies that the risk factors for resistance are the reason for the difference in breakpoint changes.  For example, the title of the section is "Impact of Resistance Factors on MIC", and while the described risk factors may have an impact on the MIC, this is not actually what you are showing with your study.  You are showing that the patients with these risk factors will be affected most with the new breakpoints.  While the breakpoint changes are independent of the risk factors you describe, because the breakpoints were already decided upon, it is still a notable finding that there are specific groups that will be impacted the most with these updated breakpoints.  These are the groups that we will be able to target our Stewardship interventions first.

  • This section of the results was updated to “Impact of Reappraisal of MIC Values among Patients with Resistance Risk Factors” to better capture this well-made point. We agree this better encompasses these data and intended discussion points.

Lastly, the last sentence in Line 229 thru 231, seems to belong more in the discussion section. 

  • We agree that this sentence (“Our data also indicates possible benefit from an antimicrobial stewardship perspective in those in nursing homes or have had recent fluoroquinolone exposure.”) was not well-placed in the statistical analysis section. We have shifted this sentence to the conclusions section.

Reviewer 2 Report

The manuscript"impact of reappraisal of Fluoroquinolone Minimum Inhibitory Concentration Susceptibility Breakpoints in Gram-Negative Bloodstream Isolates" in a summary gave us information on impacts and the reasons for these impact after reevaluated MIC susceptibility breakpoints for Enterobacteriaceae and non-fermenting gram-negative bacilli, as well as the suggestions for adopting these updated MIC breakpoint values in clinic. Overall, it is well-logical study in a clear organized manner. The manuscript has some significances and guidance for the clinic antibiotic treatment in a certain way. I recommend its publication in antibiotics journal.

Author Response

The manuscript"impact of reappraisal of Fluoroquinolone Minimum Inhibitory Concentration Susceptibility Breakpoints in Gram-Negative Bloodstream Isolates" in a summary gave us information on impacts and the reasons for these impact after reevaluated MIC susceptibility breakpoints for Enterobacteriaceae and non-fermenting gram-negative bacilli, as well as the suggestions for adopting these updated MIC breakpoint values in clinic. Overall, it is well-logical study in a clear organized manner. The manuscript has some significances and guidance for the clinic antibiotic treatment in a certain way. I recommend its publication in antibiotics journal.

  • Thank you for your feedback and comments. 

Reviewer 3 Report

Dear Authors,

The manuscript " Impact of Reappraisal of Fluoroquinolone Minimum Inhibitory Concentration Susceptibility Breakpoints in Gram-Negative Bloodstream Isolates" is very interesting and on topic.  Congratulations.

Author Response

The manuscript " Impact of Reappraisal of Fluoroquinolone Minimum Inhibitory Concentration Susceptibility Breakpoints in Gram-Negative Bloodstream Isolates" is very interesting and on topic.  Congratulations.

  • Thank you for your feedback and comments. 

Reviewer 4 Report

Fluoroquinolones belong to chemotherapeutics, which due to the wide spectrum of activity, quite good penetration into most tissues, relatively low toxicity, the possibility of oral and parenteral administration, are very often used in the treatment of infections, both in outpatient and hospital settings. A problem in the widespread use of this group of drugs is the increasing incidence of many species of bacteria resistant to fluoroquinolones in humans. The study evaluates the sensitivity of Enterobacteriacea and non-fermenting bacteria to ciproflxacin after lowering the minimum inhibitory concentrations and susceptibility breakpoints for these groups of bacteria.

Comments and questions:

1/ Keywords: bacteremia; levofloxacin; moxifloxacin; antibiotics; Escherichia coli; Acinetobacter baumannii; sepsis

The text does not mention Acinetobacter baumannii, levofloxacin, or moxifloxacin. Maybe it will be better in this form:

Keywords: bacteremia; ciprofloxacin; chemotherapeutics; Escherichia coli; non-fermenters / Pseudomonas aeruginosa

2/ In the text, non-fermenters are sometimes uppercase and sometimes lowercase - please standardize

3/ What criteria were used to classify the isolates to the genus/species? It would be good to mention this in the materials and methods

4/ what method was used to determine the MIC value? It would be good to give it in materials and methods

5/ Figure 4 - there is no explanation for the 5th column (I assume it should be - ,,none")

6/ in all figures, the ordinate should have a description - in my opinion - "Percentage of isolates"

7/ Table 2 - in the description, there should be Patients, not Ptients

8/ Table 2 In the case of analysis of the impact of resistance risk factors on MIC, a greater number of patients should be taken under consideration and, in my opinion, the compared groups should include the same or similar numbers of patients.

9/ When we give the name of the bacterium for the first time, we should use the full name, e.g. Escherichia coli. The text should be abbreviated later in the text, e.g. E. coli.

10/ Family, genus, species should be italicized – should be, ,,Enterobacteriaceae” instead of ,,Enterobacteriaceae”

Author Response

Fluoroquinolones belong to chemotherapeutics, which due to the wide spectrum of activity, quite good penetration into most tissues, relatively low toxicity, the possibility of oral and parenteral administration, are very often used in the treatment of infections, both in outpatient and hospital settings. A problem in the widespread use of this group of drugs is the increasing incidence of many species of bacteria resistant to fluoroquinolones in humans. The study evaluates the sensitivity of Enterobacteriacea and non-fermenting bacteria to ciproflxacin after lowering the minimum inhibitory concentrations and susceptibility breakpoints for these groups of bacteria.

Comments and questions:

1/ Keywords: bacteremia; levofloxacin; moxifloxacin; antibiotics; Escherichia coliAcinetobacter baumannii; sepsis

The text does not mention Acinetobacter baumannii, levofloxacin, or moxifloxacin. Maybe it will be better in this form:

Keywords: bacteremia; ciprofloxacin; chemotherapeutics; Escherichia coli; non-fermenters / Pseudomonas aeruginosa

  • We agree that the keywords section needs improvement. We have updated the keywords to include the following terms: sepsis; bacteremia; ciprofloxacin; chemotherapeutics; Escherichia coli; non-fermenters / Pseudomonas aeruginosa

2/ In the text, non-fermenters are sometimes uppercase and sometimes lowercase - please standardize

  • Thank you for this comment. Non-fermenters (lowercase) has been standardized for this term throughout the text.

3/ What criteria were used to classify the isolates to the genus/species? It would be good to mention this in the materials and methods

  • This lab uses matrix assisted laser desorption ionization time of flight mass spectrometry (MALDI-TOF) for identification of genus/species. The text was updated to include this information (Lines 215-217)

4/ what method was used to determine the MIC value? It would be good to give it in materials and methods

  • This lab uses Vitek ® instruments to determine MIC values. The text was updated to reflect this information. (Lines 217 – 219)

5/ Figure 4 - there is no explanation for the 5th column (I assume it should be - ,,none")

  • Legend updated, red to reflect “none” risk factor category

6/ in all figures, the ordinate should have a description - in my opinion - "Percentage of isolates"

  • We appreciate this comment. Figures 2-4 have been updated to reflect “percentage of isolates” as Y-axis title.

7/ Table 2 - in the description, there should be Patients, not Ptients

  • Thank you for this comment. The text has been updated to reflect this.

8/ Table 2 In the case of analysis of the impact of resistance risk factors on MIC, a greater number of patients should be taken under consideration and, in my opinion, the compared groups should include the same or similar numbers of patients.

  • Thank you for this comment. The section title has been updated to better reflect the content of the section and the results reported.

9/ When we give the name of the bacterium for the first time, we should use the full name, e.g. Escherichia coli. The text should be abbreviated later in the text, e.g. E. coli.

  • Thank you for this comment. The text has been updated to clarify Escherichia coli upon first appearance (Line 73), and abbreviated throughout the remainder of the text.

10/ Family, genus, species should be italicized – should be, ,,Enterobacteriaceae” instead of ,,Enterobacteriaceae”

  • Thank you for this comment. All appearances of Enterobacteriaceae has been updated to be italicized.